

# *Aeromonas hydrophila* infection induces Toll-like receptor 2 (*tlr2*) and associated downstream signaling in Indian catfish, *Clarias magur* (Hamilton, 1822)

Chinmayee Muduli[1], Anutosh Paria[1], Ranjana Srivastava[1], Gaurav Rathore[1] and Kuldeep K. Lal[2]

[1] Fish Health Management and Exotics Division, National Bureau of Fish Genetic Resources, Lucknow, Uttar Pradesh, India
[2] Fish Conservation Division, National Bureau of Fish Genetic Resources, Lucknow, Uttar Pradesh, India

## ABSTRACT

Motile *Aeromonas* septicaemia (MAS), caused by *Aeromonas hydrophila*, is one of the most significant bacterial disease responsible for mortality in Indian catfish, *Clarias magur*, a potential aquaculture species in the Indian subcontinent. In fish, innate immunity elicited by pathogen recognition receptors (PRRs) plays an important role in providing protection against bacterial infection. Information on PRRs including Toll-like receptors (*tlrs*) and their response to bacterial pathogens remains unexplored in magur. Toll-like receptor 2 (*tlr2*), a phylogenetically conserved germ-line encoded PRR recognizes specific microbial structure and trigger MyD88-dependent signaling pathway to induce release of various cytokines responsible for innate immune response. In the present study, *tlr2* gene of magur was characterized and downstream signaling was studied following challenge with *A. hydrophila*. The full-length cDNA of *magur tlr2* (*mtlr2*) comprised of 3,066 bp with a single open reading frame of 2,373 bp encoding 790 amino acids having a theoretical pI value of 6.11 and molecular weight of 90 kDa. Structurally, it comprised of signal peptide (1–42aa), one leucine-rich repeat region (LRR) at N-terminal (LRR1-NT: 50–73 aa) and C-terminal (LRR-CT: 588–608 aa), twenty LRRs in between, one trans-membrane (Tm) domain (609–631aa) followed by cytoplasmic TIR domain (670–783aa). Phylogenetically, *mtlr2* is closely related to pangasius and channel catfish. Highest basal expression of *mtlr2*, *myd88* and *il-1β* in spleen, *nf-kb* in anterior kidney was observed. Lowest basal expression of *mtlr2* in skin and *myd88*, *nf-kb* and *il-1β* in muscle was detected. Significant up-regulation of *mtlr2* and downstream expression occurred at 3, 8, 24 h post infection to *A. hydrophila* in important immune organs such as liver, spleen, intestine and kidney. These findings highlight the vital role of *tlr2* in eliciting innate immune defence against *A. hydrophila* infection.

Corresponding author
Gaurav Rathore,
Gaurav.Rathore@icar.gov.in

## INTRODUCTION

The innate immune system serves as the first line of defence against any invading pathogen. It plays the vital role in defence system in lower vertebrate like teleosts as it has a limited diversity and potency in adaptive immune system compared to the higher vertebrates (*Hikima, Jung & Aoki, 2011*; *Zhu et al., 2013*). The initiation and regulation of the innate immunity in vertebrates depends largely on different extracellular and intracellular germ line-encoded pattern recognition receptors (PRRs) (*Kawai & Akira, 2010*; *Takeuchi & Akira, 2010*). The PRRs, especially signaling PRRs, initiate the immune-signaling cascade following recognition of conserved pathogen-associated molecular pattern (PAMP) and damage-associated molecular pattern (DAMP) released during injury or infection (*Mogensen, 2009*). Among the major classes of signaling PRRs, toll-like receptors (*tlrs*) are pivotal in sensing extracellular and intracellular microbial PAMPs. Structurally, *tlrs* possesses an extracellular domain containing leucine-rich repeats (LRRs) that recognise different PAMP and a toll-interleukin 1 (*il-1*) receptor (TIR) domain for downstream signaling that guide activation of *nf-kb*, which leads to pro-inflammatory cytokines and chemokines production, complement cascade activation, different types of interferons production, dendritic cell maturation, secretory antimicrobial peptides production and stimulation of the adaptive immune system (*Mogensen & Paludan, 2005*; *Akira, Uematsu & Takeuchi, 2006*). These inflammatory signals released from *tlrs* are protective measure to ensure elimination of detrimental threat caused by infectious agents as well as to accelerate healing process.

*tlr2* as a member of *tlr* family is an important component of innate immune system. Among *tlrs*, *tlr2* is best known as the receptor for sensing several PAMPs such as peptidoglycan, lipoteichoic acid, lipoprotein, lipopeptide found in Gram-positive bacteria, lipoarabinomannan, atypical lipopolysaccharides (LPSs), phenol-soluble modulin, porins, glycolipids, poly I:C, LPS, yeast zymosan, glycosylphosphatidyl inositol from protozoan parasites, and also respond to Gram-negative bacteria *Porphyromonas gingivalis*, *Leptospira interrogans*, *Edwardseilla ictaluri*, *Aeromonas hydrophila*, *Vibrio parahaemolyticus* (*Ozinsky et al., 2000*; *Takeuchi et al., 2001*; *Liu et al., 2001*; *Baoprasertkul et al., 2007*; *Fan, Jia & Yao, 2015*; *Zhang et al., 2017*). To recognise these large panel of PAMPs, TLR2 forms homodimer with itself or heterodimer with TLR1 or TLR6 (*Buwittbeckmann et al., 2006*). In teleosts, TLR2 cooperate with TLR1 to recognise LPS, poly I:C, lipopeptide (*Zhang et al., 2014*), as LPS agonist TLR4 is lost from the genome of most fishes and TLR6 is also absent. *tlr2* is expressed on various immune cells including neutrophils, monocytes, dendric cells (*O'Mahony et al., 2008*). TLR2 pathway affects the chemotaxis, phagocytosis, and cytokine release of neutrophils, a critical effector cell against pathogen invasion (*Chena et al., 2019*). Upon binding to its cognate ligands, TLR2 then recruit the MYD88 and TIRAP (*mal*) that activate the IRAK(1 and 4)/TRAF6/IKK (α or β)/MAPKS cascade, which subsequently leads to the ubiquitination of IκBα and the activation of transcription factor NF-kb or AP-1 to induce the expression of host defense

genes and subsequent production of pro-, anti-inflammatory cytokines, chemokines, immunomodulatory molecules (*Zhang et al., 2014*). In recent years, structure and expression pattern of *tlr2* has been characterized in few important fish species including zebra fish (*Jault, Pichon & Chluba, 2004*), Japanese flounder (*Hirono et al., 2004*), channel catfish (*Baoprasertkul et al., 2007*), grouper (*Wei et al., 2011*), rohu (*Samanta et al., 2012*), Antratic teleosts (*Varriale et al., 2012*), shark (*Anandhakumar et al., 2012*), miiuy croaker (*Xu et al., 2013*), yellow croaker (*Fan, Jia & Yao, 2015*), rainbow trout (*Brietzke et al., 2016*), tongue sole (*Li & Sun, 2016*), silvery pomfret (*Gao et al., 2016*), turbot (*Zhang et al., 2016*; *Liu et al., 2016*), grass carp (*He et al., 2016*; *Liao et al., 2017*), common carp (*Fink et al., 2016*), golden pompano (*Wu et al., 2018*) and Dabry's sturgeon (*Tang et al., 2020*). These studies have provided background information on genomic structure of *tlr2* and also reported expression profile of *tlr2* upon stimulation with ligand or bacterial infection. Comprehensively, *tlr2* play a vital role in innate immune potentiation against wide array of ligand and pathogen in fish species.

Magur is an economically important fish species in South and South-East Asian countries. The freshwater air-breathing Indian catfish, *C. magur* (genetically distinct from *C. batrachus*) (*Devassy et al., 2015*) is a promising candidate species for freshwater aquaculture in India next to carp (*Sahoo et al., 2016*). Magur is prone to bacterial and fungal infection in all phase of farming due to scale-less body and many Gram-negative bacterial pathogens have been reported from magur includes, *A. hydrophila*, *A. veronii*, *Flavobacterium* spp, *Edwardsiella tarda*, *E. ictaluri*, *Pseudomonas* spp (*Monir et al., 2017*; *Sharma et al., 2017*; *Sharma et al., 2018*). There are no reports on Gram-positive bacterium causing disease in Indian magur. Among the diseases caused by Gram-negative bacterial pathogens, Motile *Aeromonas* septicaemia (MAS), caused by *A. hydrophila*, is one of the major causes of concern for finfish culture including magur catfish culture in the subtropical climatic condition like India. Different life stages of magur are affected by *A. hydrophila* infection and the mortality may go up to 70%–80% during rearing stages whereas 50% mortality has been recorded in grow-out ponds (*Sinha et al., 2014*; *Sharma et al., 2018*).

In magur cat fish, information on PRRs including *tlrs* is miserably inadequate and their role in eliciting innate immune response to bacterial pathogens remains unexplored. *tlr2* is one of the important PRRs and plays a significant role in recognition and immunopotentiation against bacterial pathogens. Therefore, the present study was aimed to generate full-length nucleotide sequence of magur *tlr2* (*mtlr2*) cDNA and study its expression along with its downstream signaling molecules such as *myd88*, *nf-kb* and *il-1β* in different tissues following *A. hydrophila* infection. In the context of the planned experiment, it is worth noting that the cell wall of *S. aureus* contains abundant quantity of PGN, which is a known ligand of *tlr2*. Hence, *Staphylococcus aureus* was used as positive control to validate the induction of *tlr2*. The findings of this study would help in understanding the potential role of *tlr2* in mediating innate immunity against *A. hydrophila* infection, so that control strategies can be developed.

## MATERIALS AND METHODS

### Ethical statement

This research work was carried out in compliance with the animal welfare laws in India. Guidelines of CPCSEA [(Committee for the Purpose of Control and Supervision of Experiments on Animals), Ministry of Fisheries, Animal Husbandry and Dairying, Department of Animal Husbandry and Dairying, Government of India] on care, treatment and experimental protocol of animals in scientific experimentation on fishes were followed. The animal care and experimental challenge protocol was approved by the Institutional Animal Ethics Committee (NBFGR/IAEC/2019/007) of ICAR-National Bureau of Fish Genetic Resources, Lucknow, India.

### Experimental fish and maintenance

Apparently healthy farm bred juvenile of *C. magur* ($n$ = 120; size-22.31 ± 1.34 cm; weight 50 ± 4.54 g) were handpicked after drying of magur rearing ponds in ICAR-NBFGR, Lucknow, Uttar Pradesh, India. Fishes were transported and stocked in lab condition in 2,000 L capacity fiber reinforced plastic (FRP) tank with sandy-clay bed provided with hideouts to avoid cannibalism. In each tank, 40 numbers of fish were kept. Fish were fed with commercially available magur feed (Growel Feed Pvt. Ltd, Lucknow, Uttar Pradesh, India) @ 2% of their body weight twice a day. Fishes were maintained for 1 month before carrying out the experiment. Half of the water was exchanged weekly.

### Cloning of magur *tlr2* ( *mtlr2*)

Initially, to amplify partial cDNA sequence of *tlr2* gene, PCR primers (Table 1) were designed from conserved region of the nucleotide sequences of channel catfish, *Ictalurus punctatus* (GenBank accession HQ677714.1) and yellow catfish, *Tachysurus fulvidraco* (KU950711.1) *tlr2* gene. The initial partial fragment of *mtlr2* was amplified from the spleen cDNA of *C. magur*. The amplified PCR fragment was cloned into pTZ57R/T vector (Thermo Scientific). The recombinant plasmids containing the PCR amplified fragment was sequenced in both the direction using M13 forward and reverse sequencing primers. Likewise, primers were designed from the overlapping sequences of the newly generated sequences and PCR products were sequenced for obtaining the intermittent sequences. To amplify the full-length *mtlr2* cDNA sequence, 5′ and 3′-cDNA ends were amplified using SMARTer RACE cDNA amplification kit (Clontech, San Jose, CA, USA) and the specific primers designed from the newly obtained sequences. The amplified PCR products of the cDNA ends were cloned and sequenced as mentioned earlier.

### Bioinformatics analyses of *mtlr2* sequence

The sequences of different amplicons amplified in PCR were read by Chromas version 2.6.5 and joined together based on consensus sequences to obtain the complete cDNA sequence of *mtlr2*. NCBI-BLAST (https://blast.ncbi.nlm.nih.gov/Blast.cgi) search was carried out for obtaining the sequence homology of *mtlr2* with that of other reported *tlr2*. The open reading frame (ORF) and the corresponding protein sequence were identified with the help of ORF finder tool of NCBI (https://www.ncbi.nlm.nih.gov/orffinder/).

**Table 1 Primers, their sequences and their application in this study.**

| Primers | Sequence (5′-3′) | GenBank accession no | Usages |
|---|---|---|---|
| **(A)Primers used for PCR amplification and cloning of magur TLR2** | | | |
| *tlr2*-F1 (2F) | CTGAAACATCTGAACATCTTGG | | First round homology PCR |
| *tlr2*-R1 (3R) | ACGCTAACTGTATCATGCTGC | | |
| *tlr2*-F2 (3F) | CCAGGGCTTCTTCAAATTTAGC | | First round homology PCR |
| *tlr2*-R2 (5R) | TGGACTCGATGATGTTGTCG | | |
| *tlr2*-F3 | CTTCCAGACGATTCAACCCAT | | Nested PCR |
| *tlr2*-R3 | GCTCATGTCTAGCGATGTCAG | | |
| T2-5′MR | ACGGCTCTGATGGGTTGAATCGTCTGG | | 5′ RACE |
| T2-5′MNR | AACGTTCTCAGCGCGGGGTTTTCGAC | | 5′ Nested RACE |
| T2-3′MR | GGGTCACCTGCCACAAATTCCACATC | | 3′ RACE |
| T2-3′MNR | GCTTTTCGTGTCGTACAGCCAGCAC | | 3′ Nested RACE |
| UPM | Long/CTAATACGACTCACTATAGGGCAAGCAGTGGTATCAACGCAGAGT Short/CTAATACGACTCACTATAGGGC | | |
| NUP | AAGCAGTGGTATCAACGCAGAGT | | |
| **(B)Primers used for real-time gene expression analysis** | | | |
| *tlr2*rt-F | CCAGGGCTTCTTAAATTTAGC | MT625141 | RealtimePCR |
| *tlr2*rt-R | AGGAGGTTTTGGCTAAGTCC | | |
| *ef-1α*F | GCAGCTTATCGTTGGAGTCA | AB916539.1 | |
| *ef-1α*R | GAAATTGGGACGAAAGCAACG | | |
| *myd88*-F | CTGAAGCTGTGCGTGTTTGA | JQ990986.1 | |
| *myd88*-R | CTGGAAGTCACAAGCATCGC | | |
| *nf-kb*-F | AGAAGAAGGATGTGGCCGAA | MG571500.1 | |
| *nf-kb*-R | CATGGTGATGGTTGCTTGGA | | |
| *il-1β*-F | GACCAGGAGCTCTTCAATATCT | JQ309137.1 | |
| *il-1β*-R | AGCCCTTTTAGACTTGTCACA | | |

Multiple sequence alignment of deduced amino acid sequence of *mtlr2* with vertebrates' counterparts such as human, zebra fish, common carp, rohu, channel catfish and large yellow croaker was carried out using Clustal Omega (https://www.ebi.ac.uk/Tools/msa/clustalo/) to identify the conserved features and domains. Signal peptide, transmembrane domain and TIR domain were predicted using the SMART software (http://smart.embl-heidelberg.de/) whilst the remaining LRR motifs were identified manually. LRR domain was also searched in the deduced amino acid sequence for the presence of LxxLxLxxNxL as decribed by *Matsushima et al. (2007)*. The theoretical isoelectric point ($P_i$) and molecular weight ($M_w$) were computed with protParamExpasy programme (http://web.expasy.org/protparam/). The amino acids were deduced based on cDNA sequences using the translate program found on the ExPASy website. Microsatellites in the *mtlr2* sequence were identified using microsatellite repeats finder (http://insilico.ehu.es/mini_tools/microsatellites/). Nucleotide and protein blast was carried out to determine identity percentage of *mtlr2* with other related fish species. To construct phylogenetic tree,

multiple-sequence alignment of *tlr2* amino acid sequences of fishes, human, animals was carried out with the help of Clustal Omega. Phylogenetic tree of *tlr2* was constructed by neighbour joining method of MEGA 7.0 programme bootstrapped 1,000 times using default parameters.

## Bacterial strains

Our previously isolated *Aeromonas hydrophila* 9C strain (*Muduli et al., 2020*) was used as test strain, while *Staphylococcus aureus* (ATCC 25923) was used as positive control for this experiment. The bacteria were cultured in tryptic-soy broth (TSB) at 37 °C, overnight and total plate count was estimated individually for both the strains before *in-vivo* challenge.

## *In vivo* bacterial challenge and sample collection

One hundred twenty numbers ($n = 120$) of apparently healthy magur (~50g) were randomly divided into three groups *i.e.*, negative control (Phosphate buffer saline or PBS injected), treatment (*A. hydrophila* injected) and positive control (*S. aureus* injected). Each group consisted of 40 fish in 1,000 L FRP tank. Completely randomized design (CRD) was followed to set up the experiment. Before initiation of the experiment, three fish were sampled randomly for bacterial and parasitological examination to ensure that fish were free of bacterial and parasitic infection. During the experimental period, the temperature, dissolved oxygen and pH were $35 \pm 2$ °C, $6.8 \pm 0.78$ mg L$^{-1}$, $8.2 \pm 0.45$, respectively as measured by a multiparameter water quality meter (Hanna Instruments, Nuşfalău, Romania), whereas, nitrite and ammonia were estimated to be $0.013 \pm 0.007$ mg L$^{-1}$ and $0.106 \pm 0.02$ mg L$^{-1}$, respectively following *APHA (1998)* protocol.

For bacterial infection, fishes in treatment group were intra-peritoneally (i.p.) injected with Lethal dose (LD$_{50}$) of *A. hydrophila* at $1 \times 10^5$ cfu/fish in 100μl of sterile phosphate buffer saline (PBS). Fishes in positive control were intra-peritoneally injected with *S. aureus* at $1 \times 10^5$ cfu/fish in 100 μl PBS. The negative control group were injected with100 μl of sterile PBS and kept separately. Thereafter, at 3, 8, 24, 72, 144 h post-infection (hpi), three fish from bacteria-injected and control group were randomly picked, euthanized in 300mg/L concentration of MS-222 and sacrificed. Tissues such as gill, liver, spleen, intestine and kidney tissue were dissected out in to RNAlater and stored at −80 °C until RNA extraction. At the end of the experiment, the survived fishes in *A. hydrophila* and *S. aureus* group were kept separately for further studies.

## Total RNA extraction and cDNA synthesis

Total RNA was extracted from the tissues using TRI Reagent (Sigma-Aldrich) as per the manufacturer's instructions. Tissue sample of 30–50 mg was homogenized in 1 ml of TRI reagent using lysing matrix D beads (1.4 mm) in FastPrep homogenization system (MP Biomedicals). RNA pellet was dissolved in diethyl pyrocarbonate (DEPC) treated nuclease-free water and stored at −80 °C until further use. RNA concentration and purity were analysed in Denovix UV-V is spectrophotometer (Denovix). To remove the residual genomic DNA, 1 μg total RNA was treated with 1U of DNase I, RNase free

(Thermo Scientific) following manufacturer's protocol. DNase-treated total RNA (1 μg) was reverse-transcribed using RevertAid First Strand cDNA Synthesis kit with oligo-dT primer (Thermo Scientific) as per the manufacturer's instructions.

## Quantitative real-time PCR (qRT-PCR) primers

The real-time PCR primers were designed for *mtlr2*, *myd88*, *nf-kb*, *IL-1β* and *ef-1α*. The details of primers used in real-time PCR were listed in Table 1. Before performing qPCR, amplification efficiency was determined. A series of four-fold dilution of cDNA ranging from 2,000 ng to 7.8 ng was prepared using cDNA synthesized from pooled spleen tissue of three healthy *C. magur*. Standard curve, correlation coefficient ($R^2$) and PCR amplification efficiencies (*E*) were automatically generated in the ABI StepOne Plus qPCR system based on slope of standard curve produced using four-fold diluted cDNA as template. The formula used to calculate efficiency was as follows $E\ (\%) = (10^{-1/\text{slope}} - 1) \times 100$ (*Kubista et al., 2006*). The primers with PCR efficiencies between 90%–105% were used for relative quantification.

## Basal expression analysis of *mtlr2*, *myd88*, *nf-kb*, *il-1β*

Basal expression of *mtlr2* and its downstream signaling molecules *myd88*, *nf-kb*, *il-1β* were determined. Twelve different tissues; gill, liver, spleen, intestine, anterior kidney, posterior kidney, stomach, skin, muscle, heart, brain and accessory respiratory organ (ARO) were collected separately from three naive Indian magur (~50g). Total RNA was isolated from each sample and cDNA was prepared following the manufactures protocol. Quantitative real-time PCR as mentioned later was carried out. Tissue with lowest expression was used as calibrator with respect to which fold change expression of above genes were calculated using the $2^{-\Delta\Delta Ct}$ method (*Livak & Schmittgen, 2001*). In our earlier experiment on validation of reference genes in *C. magur*, *ef-1α* was found to be the most stable reference gene for normalization of relative gene expression by real-time reverse transcription PCR in the context of pathological conditions (*A. hydrophila* injected and PBS injected) (data unpublished). Therefore, *ef-1α* was used as an internal control.

## qRT-PCR

The qRT-PCR was performed in Applied Biosystem StepOne Plus Real-time PCR system using TB Green™ Premix Ex Taq™ II (TAKARA) following manufacture's instruction. Amplification was carried out in 12.5 μl reaction volume, comprising 6.25 μl SYBR green master mix, 0.25 μl of ROX, 0.5 μl of forward and reverse primer (5 μM each), 1 μl of cDNA (100 ng/ul) and 4 μl of nuclease-free water. Each sample was run in triplicate, which served as technical replicate under following cycling condition: initial denaturation at 95 °C for 30 s followed by 40 cycles of 94 °C for 10 s, 60 °C for 10 s, 72 °C for 10 s. This was subsequently followed by melt-curve analysis for 15 s at 95 °C and 1 min at 60 °C. A no template control in triplicate was run for each plate. The data in form of threshold cycle (Ct) for each sample was normalized with corresponding *ef-1α*. The fold-change in expression in the treatment group was compared with that of PBS-injected control group using the $2^{-\Delta\Delta Ct}$ method (*Livak & Schmittgen, 2001*). The data obtained from

qRT-PCR analysis from nine technical replicate out of three biological replicate were subjected to statistical analysis.

## Statistical analysis

All data were expressed as mean ± standard error of mean (S.E.M). Statistical analysis was performed using SPSS 16.0 software (SPSS Inc., Chicago, IL, USA). The significant difference in fold change expression at different time point within individual tissue was determined with time point as independent variable and fold change in expression as dependent variable at 95% significance level using statistical model ANOVA with Tukey post-hoc test ($p < 0.05$). The $p$ values significant at <0.05, <0.01, <0.001 levels, respectively, compared to controls indicated by different superscripts.

## RESULTS

### Characterization of full-length cDNA of *mtlr2*

Full-length cDNA encoding *mtlr2* was amplified from magur spleen cDNA. Initially, two set of primer (2F-3R, 3F-5R primer) yielded 626 bp (Fragment 1) and 1,129 bp (Fragment 2) amplification. Further, a new set of internal/nested primer (F3-R3) generated 694 bp (Fr3) overlapping Fragment 1 and 2. Joining these fragments together a long stretch of 1,830 bp product was obtained. Cloning, sequencing, and NCBI-BLAST search of 1,830 bp cDNA had sequence similarity with other fish *tlr2* which confirmed it to be *tlr2*. The 3′-and 5′-RACE were performed to obtain 3′-and 5′-terminal regions of cDNA. The full-length *mtlr2* cDNA was 3,066 bp, including 5′ and 3′ un-translated region (UTR) of 113 and 582 bp, respectively. The predicted single ORF of *mtlr2* was 2,373 bp, located between 114–2,487 bp nucleotide and encodes 790 amino acids with a theoretical pI value of 6.11 and molecular weight 90.095 kDa. Structurally, it comprised of a signal peptide (1–42 aa), one leucine-rich repeat region (LRR) at N-terminal (LRR1-NT; 50–73 aa) and C-terminal LRR (LRR-CT; 588–608 aa), twenty LRRs in between N-and C-terminal, one trans-membrane (Tm) domain (609–631 aa) followed by one TIR domain (670–783 aa) (Figs. 1–3). The highly conserved motifs were identified in TIR domains, and wide range of diversity was observed within the LRR regions (Fig. 3).
Two microsatellites were detected in the UTR region of cDNA sequence of *mtlr2* *i.e.*, pentamer penta repeat $(GTTTT)_5$ at 69 bp of 3′UTR region and dimer octarepeat $(AC)_8$ repeat at 2,862 bp of 5′UTR region. One hexamer tri-repeat $(CTGACT)_3$ also present in coding region at 215 bp location. Two mRNA instability motifs (ATTTA) were found in the 3′-UTR, which suggest that *mtlr2* may be transiently expressed (Fig. 1). The *mtlr2* nucleotide sequence was submitted to GenBank database with GenBank ID: MT625141.

### Structural identity and phylogenetic relationship of *mtlr2* with other species

Similarity comparison of the *mtlr2* nucleic acid sequence with pangasius, channel catfish, and yellow catfish revealed identity percentage of 83.19, 79.20 and 80.69, respectively at nucleic acid level. At the deduced amino acid level, the corresponding identity % was 77.63,

```
   1 ATAGTTAAGCGAGAAAAGATTACTACAAGCAAGACTTTTATCTACTGGTGTACTTGATAATAACTACCGGTTTTGTTTTGTTTTGTTTTGTT92
  93 TTTAGTTTTGATTCTACTGTGATGCAAATGCGACACTTCAGGCTGATGTTTTCAGCTTCTCTCCAGAAACAACACACATCCAGCAGGATG182
                                   M  Q  M  R  H  F  R  L  M  F  S  A  S  L  Q  K  Q  H  T  S  S  R  M
 183 AAGGTGCCACTGGCCTTATGTATCTGTTTTAGCCTGACTCTGACTCTGACTCAGACCTCCGAGAGGACCAACCTGTAATGACTGCGACGAA272
        K  V  P  L  A  L  C  I  C  F  S  L  T  L  T  L  T  Q  T  S  E  R  P  T  C  N  D  C  D  E
 273 GATCATTTCTGCAACTGTCGTGCTAAGGACCTCCACGGTGTCCCATAGTTCCAGATGATGTTCTTTACCTAGACGTGTCCTTCAACGAG362
        D  H  F  C  N  C  R  A  K  D  L  H  G  V  P  I  V  P  D  D  V  L  Y  L  D  V  S  F  N  E
 363 ATCGAGTCCATCACTCAGAGGGATCTGACCTGCTACACAGAGCTGAGAAATTTAAAGCTGCAGAAGAACAAACTCAGCACGATCCACAAA452
        I  E  S  I  T  Q  R  D  L  T  C  Y  T  E  L  R  N  L  K  L  Q  K  N  K  L  S  T  I  H  K
 453 GAAGCATTTCATTCCCAAAGTAAACTGGAAGCGCTTGATCTGTCATTCAATAACCTGAAAAACATTTCCTCCCAATGGTTTTCTAATCTT542
        E  A  F  H  S  Q  S  K  L  E  A  L  D  L  S  F  N  N  L  K  N  I  S  S  Q  W  F  S  N  L
 543 CGGTCCCTGAAACATTTGAACATCTTGGGAAACCAGTACACCACTTTGGGATCCATCGCCTTGTTTCAATTTGTCGAAAACCCCGCGCTG632
        R  S  L  K  H  L  N  I  L  G  N  Q  Y  T  T  L  G  S  I  A  L  F  Q  F  V  E  N  P  A  L
 633 AGAACGTTACAGTTCGGCAACCTTTGGATCAGGGATGTGAAACAGAATTTGCTGCGTAATATTAGACAGCTGGATGAGCTGTCGTTTGTC722
        R  T  L  Q  F  G  N  L  W  I  R  D  V  K  Q  N  L  L  R  N  I  R  Q  L  D  E  L  S  F  V
 723 GGTGGTGTCCTCAGATCATATGAGAATGGAAGCTTCCAGACGATTCAACCCATCAGACCCGTGTCAGTCAGCCTTTCGCGGTTGTTTCAG812
        G  G  V  L  R  S  Y  E  N  G  S  F  Q  T  I  Q  P  I  R  A  V  S  V  S  L  S  R  L  F  Q
 813 GATGATCCAGCACTGGTATCAAAGATCCTTCGAGATGTTTCTCACCCTGAGACATCGCTGACCATTAGAGATGTCTCCCTGGAGACACAA902
        D  D  P  A  L  V  S  K  I  L  R  D  V  S  H  P  E  T  S  L  T  I  R  D  V  S  L  E  T  Q
 903 GAACTGATAGAACCCTTAAGAGAGGTGACAGAAGGTGGCACCAGAAGTCTTACCTTTCAAAACATAATCACAACTGACGAGGCAGTCAGC992
        E  L  I  E  P  L  R  E  V  T  E  G  G  T  R  S  L  T  F  Q  N  I  I  T  T  D  E  A  V  S
 993 CGCCTTCTGGAGGTTTTGGACGGCTCTCCGGTGTCCTACATCGGCCTTGAGGACATTTGTTTAATAGGTCAGGGCTGGTGGGAAAAGGCG1082
        R  L  L  E  V  L  D  G  S  P  V  S  Y  I  G  L  E  D  I  C  L  I  G  Q  G  W  W  E  K  A
1083 AAGAGGACACACCTAGAAAACCTGCACACGATACATGTCCGCAACATAGAAATCCAGGGCTTCTTCAAATTTAGCAGCATGATACAGTTA1172
        K  R  T  H  L  E  N  L  H  T  I  H  V  R  N  I  E  I  Q  G  F  F  K  F  S  S  M  I  Q  L
1173 GCGTTCCTGTTGAAGCACCTCACCAAGATATCCGTCATCAACTGCACCGTTTTCGTTATTCCCTGCCTGACCAGCTGTTTTCTTAAAAAG1262
        A  F  L  L  K  H  L  T  K  I  S  V  I  N  C  T  V  F  V  I  P  C  L  T  S  C  F  L  K  K
1263 GTGGAGTACTTGGACTTGAGCCAAAACCTCCTCTCGGATATCACCATGCAAGAATCCCTGTGCAACGGGGACAGCAAGATGCGCAATATT1352
        V  E  Y  L  D  L  S  Q  N  L  L  S  D  I  T  M  Q  E  S  L  C  N  G  D  S  K  M  R  N  I
1353 AACACGCTCAATGTAAGTCACAACTCGCTGAAATCTCTGCAGCTCATGTCCCACCTGGTCACGAGTCTCGACAGGCTGACATCGCTAGAC1442
        N  T  L  N  V  S  H  N  S  L  K  S  L  Q  L  M  S  H  L  V  T  S  L  D  R  L  T  S  L  D
1443 ATGAGCCACAACAACTTTGTAAAGATGCCACAGAGTTGCAGCTGGCCGGCAAGTCTCAGGTTTATGAACCTGTCCACTACAAAACTTCAC1532
        M  S  H  N  N  F  V  K  M  P  Q  S  C  S  W  P  A  S  L  R  F  M  N  L  S  T  T  K  L  H
1533 CGCGTAACCCCGTGCCTACCTCTCAGCCTGACCGTGCTGGATTTGAGCCAGAACTTCCTGACAGAGTTCCACCTCCATCTTCCCAACCTT1662
        R  V  T  P  C  L  P  L  S  L  T  V  L  D  L  S  Q  N  F  L  T  E  F  H  L  H  L  P  N  L
1663 GCGGGAGCTCTGGCTTACAGGGAACAGGATTATTGCCCTGCCGGAAGGTGGCCACTTCCCCAGCCTACGCATGCTGTTTATTCAAAGCAAC1712
        A  E  L  W  L  T  G  N  R  I  I  A  L  P  E  G  G  H  F  P  S  L  R  M  L  F  I  Q  S  N
1713 CTGACCATGTTCAACAAAAGCGACCTGATGGCGTTCCAGTCTCTCCAGGTCTTGGAAGCCGACATAACAATTTTTTTTGCAGCTGC1802
        T  L  N  M  F  N  K  S  D  L  M  A  F  Q  S  L  Q  V  L  E  A  G  H  N  N  F  F  C  S  C
1803 GATTTCGTAGAATTCTTTCAAGGTTCTATTGACCACTTGATCACTCTGGGGGACGGACATCGCAGCTACATGTGTGACTCTCCGTTCACG1892
        D  F  V  E  F  F  Q  G  S  I  D  H  L  I  T  L  G  D  G  H  R  S  Y  M  C  D  S  P  F  T
1893 TTAAGGGGTCTTAATATAGATACCGCTCAACCGCCAGTCTTCGAGTGCTACATGATCCTGTTAGTATCAGTCATCTGCTCGGTCACCGTC1982
        L  R  G  L  N  I  D  T  A  Q  P  P  V  F  E  C  Y  M  I  L  L  V  S  V  I  C  S  V  T  V
1983 ATCGGCGTGATCGCCATCGGGGTCACCTGCCACAAATTCCACATCTTGTGGTACCTGCAGATGATGATCGCGTGGTTAAAAGCAAAGAGT2072
        I  G  V  I  A  I  G  V  T  C  H  K  F  H  I  L  W  Y  L  Q  M  M  I  A  W  L  K  A  K  S
2073 AAACCATCCGTGCAAATGGCGGCGCTACTTTTCGATACGATGCTTTTCGTGTCGTACAGCCAGCACGATGCGCAGTGGGTGGAGGAAATC2162
        K  P  S  V  Q  M  A  A  L  L  F  D  T  M  L  F  V  S  Y  S  Q  H  D  A  Q  W  V  E  E  I
2163 CTCGTGCCCAGAGTTAAAAAGCTCTGAGTCTCCGCTCGCTCTGTGTCTGCACAGCGGGACTTCCTCCCAGGCCGCTGGATCGCCGACAAC2252
        L  V  P  E  L  K  S  S  E  S  P  L  A  L  C  L  H  Q  R  D  F  L  P  G  R  W  I  A  D  N
2253 ATCATCGAGTCCATCGAAAGCAGCTATCGGACCCTCTTTGTCCTGTCGGAGAACTTCGTGACGAGCGAGTGGTGCCGATACGAGCTGAAC2342
        I  I  E  S  I  E  S  S  Y  R  T  L  F  V  L  S  E  N  F  V  T  S  E  W  C  R  Y  E  L  N
2343 TTTTCGCATTTTCGGATCATCGACGAGCGCAACGATTCGGCCGTCCCTGATCCTGCTAGAGCCCATCGCCAAGGAGACGATTCCCAAGCG2432
        F  S  H  F  R  I  I  D  E  R  N  D  S  A  V  P  D  P  A  R  A  H  R  Q  G  D  D  S  Q  A
2433 CTTCTGCAAACTGCGCAAAATAATGAACTCCAGGACGTACCTCGAGTGGCCTGAGGACGAAGAAAGCGAGAGGAATTTTGGCACAATCT2522
        L  L  Q  T  A  Q  N  N  E  L  Q  D  V  P  R  V  A  -
2523 CCGAGCTGCACTTAGAAGGGAGGACTCGTGATGCCGATTCTGATACGTCACGGATCACACCTTCATCAGATTTGAAAATAATTACCCGGT2612
2613 TTCATTTTGTCAGTAATTTGAAGTTCTGACAGCGGGAGAACCCTTACTTGTTACTTGTGGTGTGTCTACAGCATACGCACTTATTTACCGT2702
2703 TAGAGCTCGTTATTCCTCACCTGATGTACAGATTAGACTCAAACCCAGACTCAACCTCTGATTAAGTTCTGCTCTCAATAAACACAAGCT2792
2793 CTATCTATTTAAGGATCTTCTGCATCGGGGGCTTTCCTTTTTATGGTTAGCTTCCTCATAAAATGGAGAACACACACACACACACATGCA2882
2883 CACATCTTATGAGTAATTCTTCAGTTATTCACTCAATGAAACCTTCCTCATGATGGTTAGATTCGTGCTGTCTTGTTTTAATATGATTTT2972
2973 ATATTACAGAAGTGGAAAGATTATTAATAATTAACTTGTGAAAATAAACGCAGCGAGGAATCATTTCAAAAAAAAAAAAAAAAAAAAAAA3062
3063 AAAA                                                                                      3066
```

**Figure 1 Nucleotide and deduced amino acid sequence of *mtlr2* cDNA.** Start codon and stop codon are marked in magenta colour. In the deduced amino acid sequence, signal peptide (1–42 aa) is highlighted by magenta colour. LRR-NT and LRR-CT are highlighted in grey, LRRs are highlighted in yellow, transmembrane domain highlighted in turquoise and the cytoplasmic TIR domain is highlighted in olive green. Extracellular LRR-NT (50–73 aa), LRR1 (98–115 aa), LRR2 (122–139 aa), LRR3 (143–164 aa), LRR4 (173–187 aa), LRR5 (173–459 aa), LRR6 (197–215 aa), LRR7 (228–239 aa), LRR8 (250–268 aa), LRR9 (277–296 aa), LRR10 (299–316 aa), LRR11(331–347 aa), LRR12 (360–373 aa), LRR13 (387–401 aa), LRR14 (415–434 aa), LRR15 (439–456 aa), LRR16 (462–478 aa), LRR17 (483–498 aa), LRR18 (503–519 aa), LRR19 (525–535 aa), LRR20 (543–559 aa), LRR-CT (588–608 aa). Transmembrane domain (609–631aa), intracellular cytoplasmic TIR domain (670–783 aa). Two mRNA instability motifs (ATTTA) marked in red colour underlining.

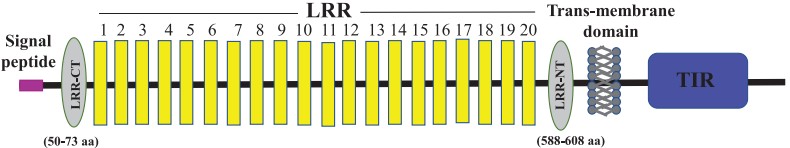

**Figure 2 Schematic representation of *C. magur tlr2* (*mtlr2*) protein domains.** Signal peptide (pink), LRR domains (yellow), LRR-CT and LRR-NT (grey) followed by transmembrane and TIR domain (blue).

76.4, and 72.83, respectively with pangasius, channel catfish, and yellow catfish. With carp and higher vertebrate family, the identity % of *mtlr2* amino acids was ~60 and ~37, respectively. The identity of conserved TIR domain of *mtlr2* ranged from 89.9% with pangasius catfish to lowest 57.4% with Japanese flounder. TIR domain of *mtlr2* shares highest identity with catfish group *i.e.* 89.9% with pangasius. TIR domain of *mtlr2* shares identity of 77.5% with zebrafish, 64% with mouse and 65.2% with human (Table 2).

Phylogenetic tree was constructed to understand the evolutionary relationship of *mtlr2* with *tlr2* reported from other teleosts. Among the fishes, *tlr2* representing catfish group including magur, channel catfish, yellow catfish and pangasius were closely related and formed one cluster, and were well separated from carps *viz.*, rohu, mrigal, common carp, grass carp and zebra fish. Within the members of catfish family, *mtlr2* branched with pangasius and channel catfish, indicating close evolutionary relationship. Other group of fishes such as trout, bream, flounder, croaker, grouper, and cod formed a separate cluster from carps and catfishes. As oblivious from phylogenetic tree that fish *tlr2* are well divergent from other higher animals (cluster-IV) (Fig. 4).

## Basal expression of *mtlr2* and its downstream molecules

The qRT-PCR amplification efficiencies ($E$) of primer sets fell within the suitable experimental range as per MIQE guideline (*Bustin et al., 2009*) *i.e.*, from 95.81% to 104.66%; while correlation coefficient ($R^2$) ranged from 0.989 to 0.998 respectively (Table 3). Tissue specific expression of *mtlr2, myd88, nf-kb* and *il-1β* gene in twelve different tissues including gill, liver, spleen, anterior kidney, posterior kidney, intestine, accessory respiratory organ (ARO), stomach, muscle, brain, heart and skin was evaluated by qRT-PCR assay. Their expression in various organs/tissues was normalized with the reference gene *ef-1α* and represented as relative fold changes from the lowest expressing tissue (calibrator). Highest basal expression of *mtlr2, myd88* and *il-1β* in spleen, *nf-kb* in anterior kidney was observed. Lowest basal expression of *mtlr2* in skin and *myd88*, *nf-kb*, *il-1β* in muscle was detected. Although, the tissue-specific expression of *mtlr2* was varied but the expression was at detectable level in all the examined tissue (Figs. 5A–5D).

## Inductive expression of *mtlr2* in response to *S. aureus*

Modulations of *mtlr2* gene transcripts were analyzed in gill, liver, spleen, intestine and kidney by qRT-PCR following i.p. injection of *S. aureus*. As shown in Fig 6, *mtlr2* expression was significantly induced in various tissues at different time points.

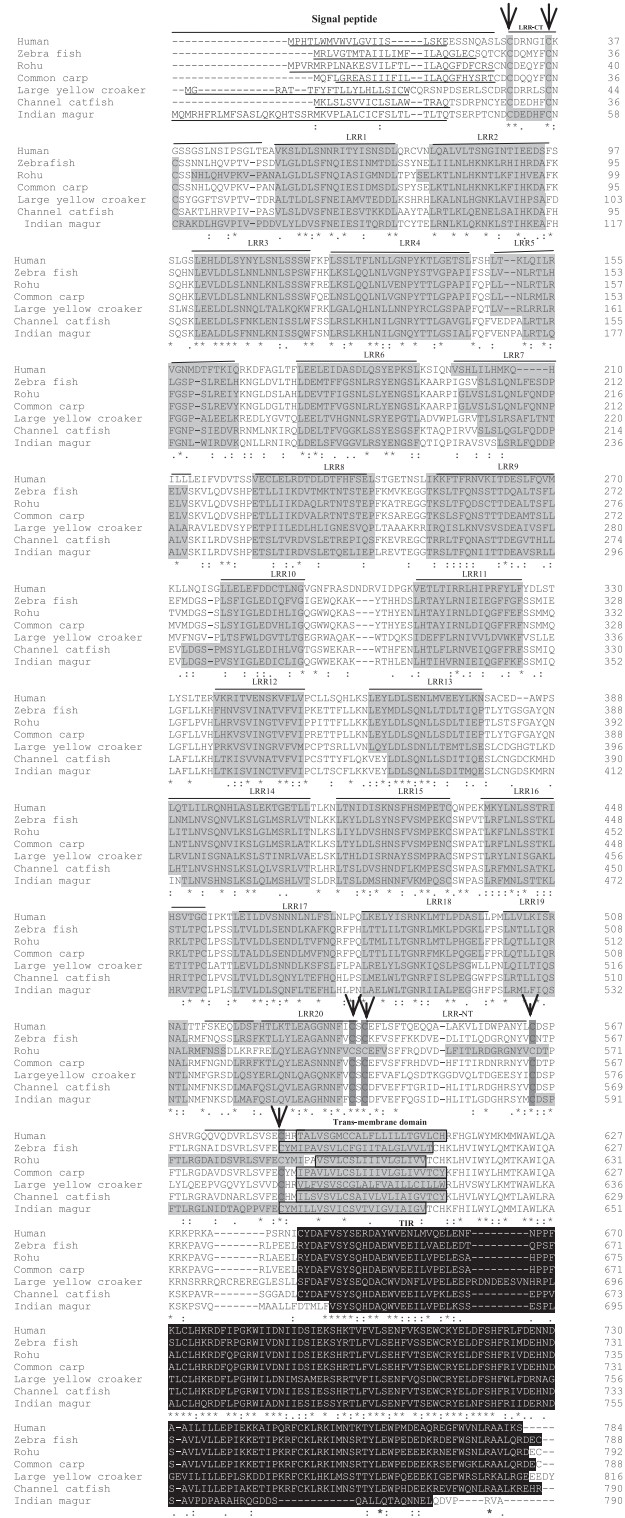

**Figure 3 Alignment of *tlr2* protein sequences.** The sequences from human, zebra fish, rohu, common carp, large yellow croaker and channel catfish were aligned with magur *tlr2* sequence to identify the conserved features and domains. Dashes represented amino acid deletions. The signal peptide is underlined, LRRs are highlighted in grey, transmembrane domain is indicated by boxes and also highlighted in grey, and the TIR domain is highlighted in black. Furthermore, LRRNT and LRRCT cysteines are indicated with arrows.

**Table 2  Identity of magur *tlr2* at nucleotide and amino acid level with other fish and animal species.**

| Species | Common name | *mtlr2* nucleotide identity (%) | *mtlr2* deduced aminoacid Identity (%) | Tir domain aminoacid Identity (%) |
|---|---|---|---|---|
| *Pangasionodon hypophthalmus* | Pangasius | 83.19 | 77.63 | 89.9 |
| *Ictalurus punctatus* | Channel catfish | 79.20 | 76.4 | 86.52 |
| *Tachysurusfulvidraco* | yellow catfish | 80.69 | 72.83 | 87.64 |
| *Pygocentrus nattereri* | red bellied piranha | 71.38 | 63.98 | 78.65 |
| *Labeo rohita* | rohu | 66.46 | 59.9 | 82.02 |
| *Cirrhinus mrigala* | mrigal | 65.76 | 60.2 | 83.15 |
| *Megalobrama amblycephala* | Wuchang bream | 66.52 | 60.4 | 80.9 |
| *Ctenopharyngodon idella* | Grass carp | 66.56 | 60.2 | 82.02 |
| *Danio rerio* | zebrafish | 58.8 | 58.1 | 77.5 |
| *Takifugu rubripes* | fugu | 45.6 | 44.1 | 59.3 |
| *Oncorhynchus mykiss* | rainbow trout | 55.0 | 48.4 | 69.7 |
| *Oreochromis niloticus* | Nile tilapia | 53.7 | 43.9 | 57.9 |
| *Paralichthys olivaceus* | Japanese flounder | 46.6 | 42.9 | 57.4 |
| *Chionodraco hamatus* | Antarctic teleost icefish | 53.7 | 46.2 | 59.4 |
| *Homo sapiens* | Human | 51.3 | 38.7 | 65.2 |
| *Gallus gallus* | Chicken | 50.1 | 37.3 | 64 |
| *Bos taurus* | Cattle | 51.4 | 38.4 | 64 |
| *Canis lupus familiaris* | Dog | 52.3 | 39 | 65.2 |
| *Oryctolagus cuniculus* | Rabbit | 51.0 | 38.4 | 65.2 |
| *Mus musculus* | Mouse | 52.5 | 37.9 | 64 |

## Differential expression of *mtlr2* and its downstream signaling molecules in response to *A. hydrophila* infection

Modulation of *mtlr2* expression was studied following *A. hydrophila* infection in gill, liver, spleen, intestine and kidney. The kinetics of up-or down-regulation of the *mtlr2* gene exhibited a similarity of general trend, rising to a high level at 3 to 24 hpi and then falling to the basal level at 72 to 144 hpi, but the up-and down-regulation differed at different time points in different tissues. A significant up-regulation of *mtlr2, myd88, nf-kb* gene expression was observed in gill, liver, spleen, intestine and kidney within 24 hpi (Figs. 7A–7C). Similarly, significantly up-regulated expression of *il-1β*, a pro-inflammatory cytokine was noticed in gill, liver, spleen, intestine and kidney within 24 hpi (Fig. 7D).

## DISCUSSION

In the present study, we have identified an orthologue of mammalian *tlr2* in Indian catfish, *C. magur* which is a potential aquaculture species in South-East Asia. Two mRNA instability motifs (ATTTA) were found in the 3′-UTR, which suggest that *mtlr2* may be transiently expressed. Two mRNA instability motifs (ATTTA) were also reported in turbot (*Zhang et al., 2016*). The full-length cDNA of *mtlr2* is of 3,066 bp and comprised of an ORF of 2,373 bp nucleotide in length, encoding 790 amino acid residues. The putative *mtlr2* protein possesses the four typical component of *tlr* families from N-to C-terminus

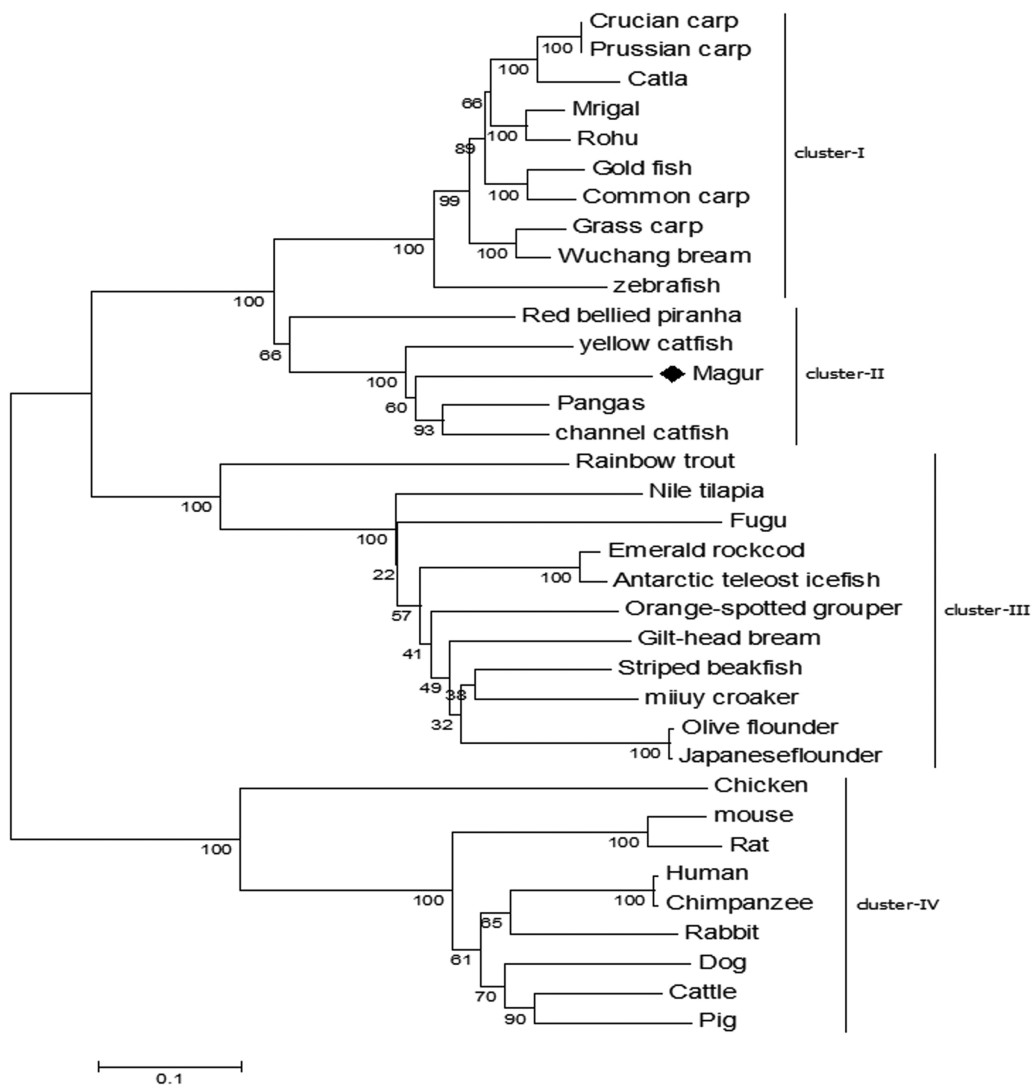

**Figure 4 Phylogenetic relationship of magur *tlr2* with other species.** *tlr2* amino acid sequences of fishes, human, animals were aligned using CLUSTALW program and the phylogenetic tree was constructed using the neighbour-joining method with the help of MEGA7 software. The branches were validated by bootstrap analysis from 1,000 replications, which were represented as bootstrap value in branch nodes. XP_026784977.1 Pangasius Catfish, AIT83001.1 gilt-head bream, ACT64128.1 Emerald rockcod, AFP44842.1 Nile tilapia, AFZ81806.1 striped beakfish, BAD01044.1 Olive flounder, AFG21856.1 miiuycroaker, CCK73195.1 rainbow trout, ADF81060.1 catla, XP_026114714.1 gold fish, AHI59129.1 Mrigal, AGO57934.1 crucian carp, AGR53440.1 prussian carp, ACT68333.1 Grass carp, ANI19836.1 Wuchang bream, XP_017572771.1 *red bellied piranha*, ANA09009.1 yellow catfish, ACT64128.1 Emerald rock cod, ACT64127.1 Antarctic teleost icefish, AEB32453.1 orange-spotted grouper, BAD01046.1 Japanese flounder, AAW69370.1 fugu, ABD17347.1 channel catfish, NP_997977.1 zebrafish, ACP20793.2 common carp, ADQ74644.1 rohu, BAB16113.2 Chicken, AAH14693.1 mouse, NP_942064.1 Rat, NP_001076250.1 Rabbit, AAH33756.1 Human, ADH84419.1 Chimpanzee, NP_001005264.2 Dog, ACH92789.1 Cattle, BAD90590.1 Pig.

*i.e.*, signal peptide, one LRR each at N-and C-terminal, twenty extracellular LRR domains in between N-and C-terminal, trans-membrane domain and highly conserved cytoplasmic TIR domain. Number of LRR domain in other fishes reported are nine LRRs in

**Table 3 The amplification efficiencies ($E$) and correlation coefficient ($R^2$) of qRT-PCR primers used in this study.**

| Symbol | Amplification efficiencies (%) | Correlation coefficient ($R^2$) |
|---|---|---|
| *tlr2* | 104.66 | 0.998 |
| *myd88* | 102.45 | 0.989 |
| *nf-kb* | 97.94 | 0.989 |
| *il-1β* | 95.81 | 0.996 |
| *ef-1α* | 99.25 | 0.998 |

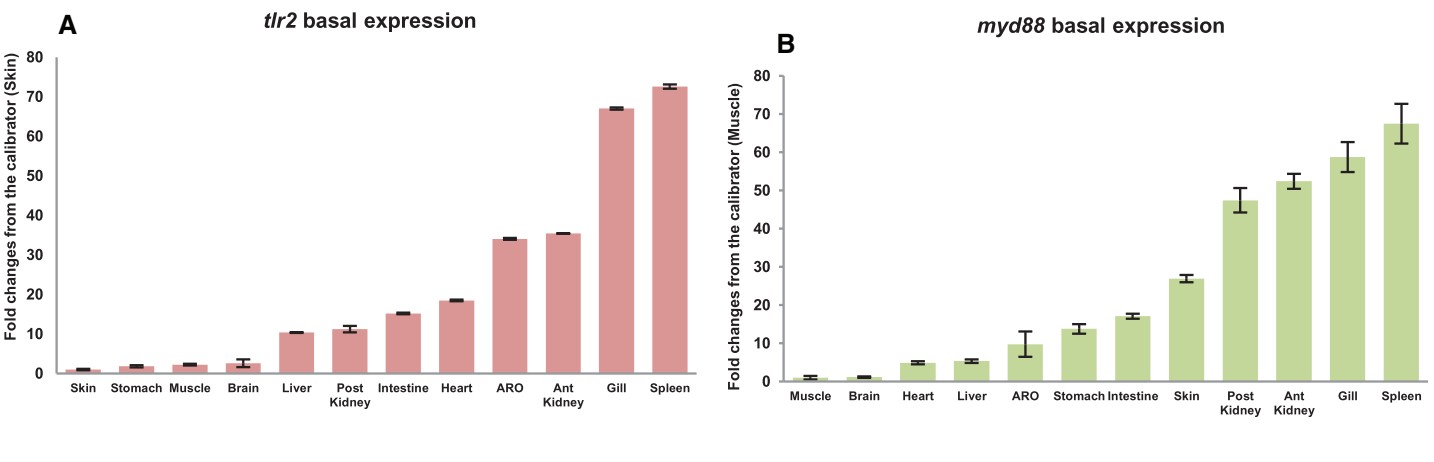

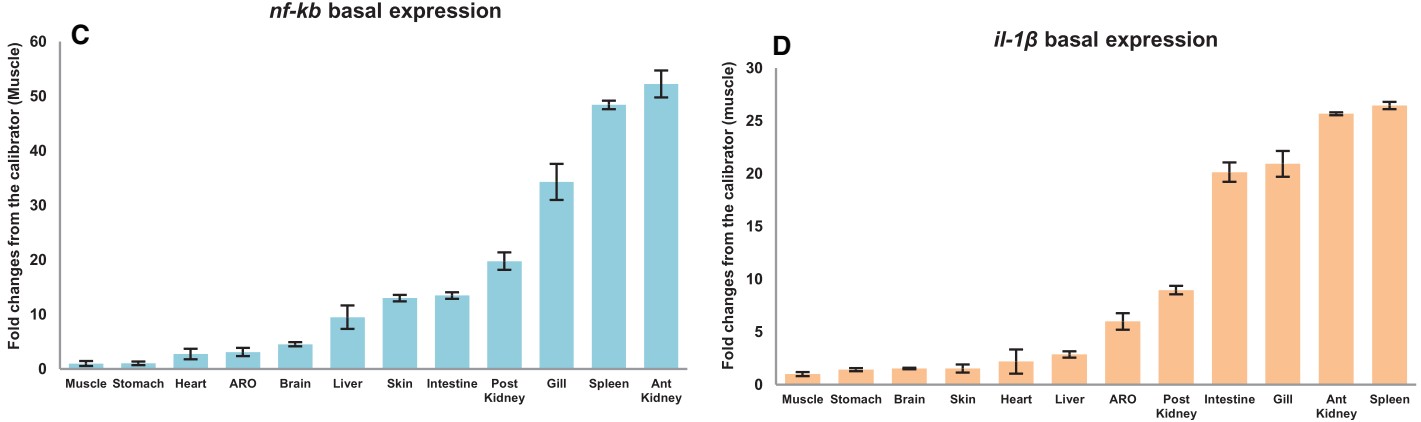

**Figure 5 Basal expression of *mtlr2* (A), *myd88* (B), *nf-kb* (C) and *il-1β* (D) genes in various tissues.** The qRT-PCR was carried out to analyse basal expression of these genes in gill, liver, spleen, intestine, anterior kidney, posterior kidney, stomach, skin, muscle, heart, brain and accessory respiratory organ (ARO). The mRNA transcript levels in various tissues were represented as ratio relative to *ef-1α* (internal control). Basal expression in the studied tissue represented as fold change from calibrator. Skin expressed lowest *mtlr2*, was used as calibrator and muscle used as calibrator for *myd88*, *nf-kb* and *il-1β*. The results were expressed as mean ± standard error of mean (bars) from three fish (*n* = 3).

channel catfish (*Baoprasertkul et al., 2007*), 21 in rohu (*Samanta et al., 2012*), 20 LRRs in human, zebra fish, channel catfish, common carp (*Fink et al., 2016*), 19 in turbot (*Zhang et al., 2016*), 11 in silvery pomfret (*Gao et al., 2016*), 8 in yellow catfish (*Pelteobagrus*

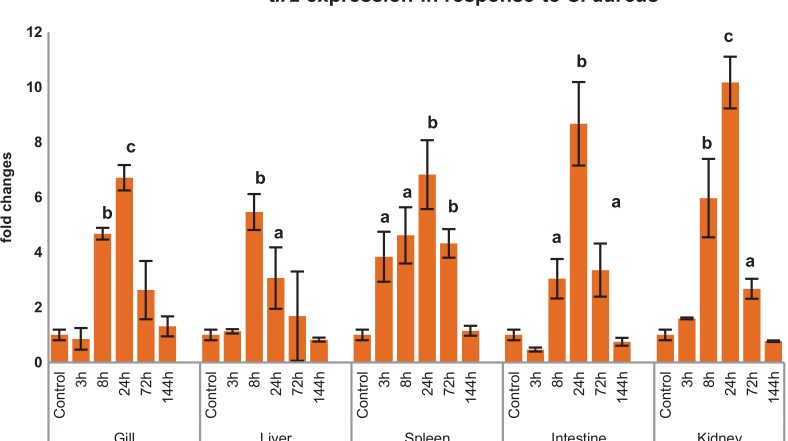

**Figure 6 Modulation of *mtlr2* expression following *S. aureus* infection.** *S. aureus* ($1 \times 10^5$ cfu/fish) was injected into magur juveniles by intra peritoneal route. At each experimental time point, total RNA was extracted from gill, liver, spleen, intestine, kidney; qRT-PCR was conducted to analyse the expression of *mtlr2*. *ef-1α* was used as an internal control. The results were expressed as mean ± standard error (bars) from three fish ($n = 3$). [a,b,c]: *p* values significant at <0.05, <0.01, <0.001 levels, respectively, compared to controls (one-way ANOVA).

*fulvidraco*) (*Zhang et al., 2017*), 8 in Dabry's sturgeon (*Tang et al., 2020*). The number of LRR signifies the functional capability to respond various agonists. Structurally, *mtlr2* is closely related to channel catfish *tlr2* gene, which also consists of an ORF of 2,373 bp encoding protein of 790 amino acids (*Baoprasertkul et al., 2007*). Our finding on number of LRRs are similar to finding by *Samanta et al. (2012)*; *Fink et al. (2016)*; *Zhang et al. (2016)*. Also, the genomic structure of a single exon and no intron reported in human, zebra fish, and channel catfish *tlr2* (*Baoprasertkul et al., 2007*), but fugu and Japanese flounder *tlr2* gene encoded by 11 and 12 exons, respectively (*Oshiumi et al., 2003*; *Hirono et al., 2004*). Whether *mtlr2* gene possesses introns or not needs further investigation. Two microsatellites were detected in the UTR of *mtlr2* *i.e.*, pentamer pentarepeat $(GTTTT)_5$ at 69 bp of 3′UTR and dimer octarepeat $(AC)_8$ at 2,862 bp of 5′UTR. One hexamer tri-repeat $(CTGACT)_3$ also present in coding region. In channel catfish, two microsatellites were identified in *tlr2* sequence *i.e.*, $(AT)_9$ repeat at 632 bp and an $(AC)_{21}$ repeat at 3,606 bp (*Baoprasertkul et al., 2007*). The identified microsatellite will allow the mapping of *tlr2* gene and future quantitative trait loci (QTL) analysis in *C. magur*.

The phylogenetic tree constructed using different vertebrate *tlr2* with that of *mtlr2* showed a possible four major clades; carps, catfishes, other teleosts, and higher vertebrates. Phylogenetically, *mtlr2* is closely related to pangasius and channel catfish. The close phylogenetic relationship indicates the functional similarities. Study on channel catfish *tlr2* amino acid showed 60.2% sequence similarity with zebrafish and 41.1% sequence similarity with human and mouse, whereas conserved TIR domain of channel catfish identity ranged from 82.2% with zebrafish and 65.1% with flounder (*Baoprasertkul et al., 2007*). However, in our study, *mtlr2* showed 58.1% amino acid sequence similarity with zebra fish *tlr2* while only 38.7 % amino acid sequence identity with human and 37.9% with mouse. TIR domain of *mtlr2* shares highest identity with other catfish group

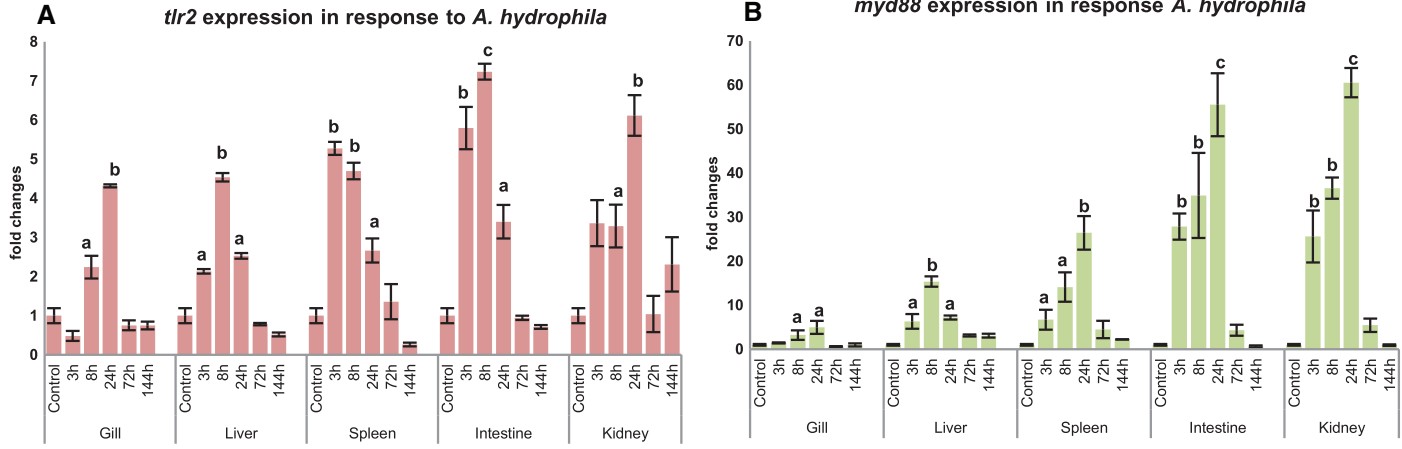

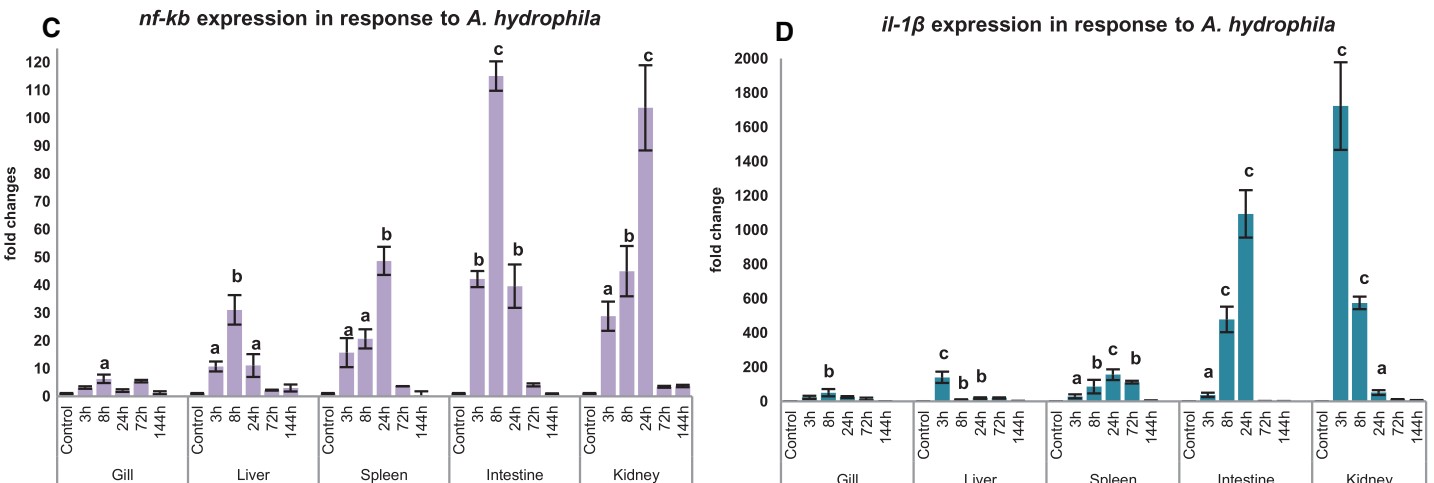

**Figure 7 Modulation of *mtlr2* (A), *myd88* (B), *nf-kb* (C), *il-1β* (D) expression following *A. hydrophila* infection.** *A. hydrophila* ($1 \times 10^5$ cfu/fish) was injected into magur juveniles by intra peritoneal route. At each experimental time point, total RNA was extracted from gill, liver, spleen, intestine, kidney; qRT-PCR was conducted to analyse the expression of *mtlr2* and its down-stream signaling molecules. *ef-1α* was used as an internal control. The results were expressed as mean ± standard error (bars) from three fish ($n = 3$).[a,b,c]: *p* values significant at <0.05, <0.01, <0.001 levels, respectively, compared to the controls (one-way ANOVA).

*i.e.,* 89.9-87.64%, 77.5% with zebrafish, 64% with mouse and 65.2% with human. The conserved TIR domain of *mtlr2* identity ranged from 89.9% with pangasius to lowest 57.4% with Japanese flounder. This indicates huge structural variability in TIR domain between fresh water and marine water fishes.

Basal expression analysis of *mtlr2* and its downstream-signaling molecule *myd88*, *nf-kb* and *il-1β* showed constitutive expression in all the twelve tissues examined. The *mtlr2* showed higher level of basal expression in spleen and lower basal expression in skin. High level of *tlr2* basal expression has been detected in liver of channel catfish, spleen of yellow catfish, spleen of rohu, liver of turbot, muscle of tongue sole, head kidney of ray fish, blood of golden pompano, head-kidney of common carp. Lowest level of *tlr2* basal expression has been detected in skin and muscle of channel catfish, eye of yellow cat fish,

eye and skin of rohu, skin of turbot, intestine and blood of tongue sole, liver of golden pompano, muscle in common carp, intestine of ray fish, (*Baoprasertkul et al., 2007*; *Samanta et al., 2012*; *Liu et al., 2016*; *Li & Sun, 2016*; *Fink et al., 2016*; *Zhang et al., 2017*; *Wu et al., 2018*; *Tang et al., 2020*). The variation in the basal expression of *tlr2* in different teleosts might be attributed to their wide diversity in habitat and the microbial invaders they encounter. In the present study, highest basal expression of *myd88* was detected in the spleen and lowest in the muscle. Basal expression analysis of *myd88* in yellow catfish (*Pelteobagrus fulvidraco*) showed ubiquitous expression with the highest *myd88* expression detected in spleen and the lowest in liver (*Yu et al., 2018*). The basal expression of *nf-kb* in magur was lowest in muscle, and highest in anterior kidney. Literature survey shows that till now there is no study on basal expression of this very important gene of TLR pathway in any fish species. This is the first report on basal expression of *nf-kb* gene in magur. Further, in this study, lowest basal expression of *il-1β* observed in muscle and calibrated higher basal expression in spleen. In large yellow croaker *lcil-1β* showed higher expression in head kidney followed by gill, liver, and intestine (*Wu et al., 2015*).

*tlr2* responds to various ligand (PGN, LTA, LPS, zymosan etc.) and pathogens in higher vertebrates as well as in lower vertebrates like teleosts. Our finding also validates that *mtlr2* is induced in response to *S. aureus* in several tissues of magur. As literature suggests that PGN is the major cell wall component of this bacterium, which might be responsible for inducing expression of *mtlr2*. Inductive expression of *mtlr2* in response to PGN of Gram-positive bacteria have been reported in large orange spotted grouper (*Wei et al., 2011*), yellow croaker (*Fan, Jia & Yao, 2015*), rohu (*Samanta et al., 2012*), common carp (*Ribeiro et al., 2010*; *Fink et al., 2016*). There are few reports on induction of fish *tlr2* to Gram-negative pathogens also. Significant up-regulation of *lctlr2* transcript in yellow croaker was detected after *V. parahaemolyticus* infection and LPS stimulation (*Fan, Jia & Yao, 2015*). Up-regulated expression of *tlr2* in response to *A. hydrophila* has been reported in channel catfish, hybrid sturgeons, yellow catfish (*Mu, Pridgeon & Klesius, 2013*; *Zhang et al., 2017*; *Wen et al., 2017*; *Jiang et al., 2018*). This could be due to binding of *tlr2* to lipoproteins, PGN carried by *A. hydrophila* (*Zhang et al., 2017*). These findings suggest that the *tlr2* in fish may also recognise Gram-negative bacteria in addition to its known ligands, but more evidence is needed to substantiate this.

Significant up-regulation of *mtlr2* detected in gill, liver, spleen, intestine and kidney at 3, 8 or 24 hpi indicating early response of *tlr2* to *A. hydrophila* infection in magur was observed in our study. These findings are supported by *Mu, Pridgeon & Klesius (2013)*, they have reported that transcriptional kinetics of the *tlr2* peaked in the anterior kidney (~5 fold increase) of channel catfish at 3 h post exposure to *A. hydrophila*. Zhang et al. (2017) also reported that *A. hydrophila* infection increased *tlr2* expression in spleen, head kidney and trunk kidney at 12 or 24 hpi. *A. hydrophila* infection challenge in mrigal induced *tlr2* expression in kidney (~4.5 fold at 12 h), gill (~1.5 fold at 12 h) and in the liver, intestine (*Basu et al., 2012*). Our result shows that, as a part of innate immune defence system, *mtlr2* expression is enhanced following *A. hydrophila* infection in magur.

In this study, a significant up-regulated *mtlr2, myd88, nf-kb, il-1β* gene expression observed in liver, spleen, intestine and kidney at 3, 8, 24 hpi, in gill at 8 or 24 hpi. *myd88* is a key adopter molecule and play an important role in facilitating downstream signaling of all toll-like receptors except *tlr3*. Our finding is supported by *Huang et al. (2018)*, they reported a significant higher level of *myd88* expression at 8 hpi in *Anguilla japonica* infected with *E. tarda* in head kidney, spleen, liver, intestine, gill and skin. In Wuchang bream (*Megalobrama amblycephala*), *myd88* expression peaked at 12 hpi of *A. hydrophila* (*Lai et al., 2017*). All the TLR pathways culminate in activation of transcription factor *nf-kb* which control expression of wide array of cytokines (*Kawai & Akira, 2007*). The effecter cytokine molecule *il-1β* play crucial role in physiological events such as immune response, inflammation, cell growth and differentiation, apoptosis. Significantly up-regulated expression of *nf-kb* and *il-1β* in the spleen and muscle has been reported in koi carp following *A. sobria* infection (*Byadgi et al., 2018*).

A well-correlated expression pattern of *mtlr2* and the downstream signaling molecules *i.e. myd88, nf-kb* and *il-1β* was observed in the respective tissues, during the initial hours (3 to 24 hpi) of infection. Higher expression of *mtlr2, myd88, nf-kb, il-1β* in internal organ such as intestine, kidney, spleen and liver than gill may be due to intra-peritoneal mode of experimental bacterial infection. Further, it also can be speculated that intestine and kidney are the two earliest responding organs to *A. hydrophila* infection. The results obtained in the present study are in agreement with the findings by *Samanta et al. (2012)* and *Li & Sun (2016)*, on regulation of *tlr2* and its downstream-signaling molecules in rohu and tongue sole, respectively following bacterial infections.

## CONCLUSION

In conclusion, full-length cDNA sequence of magur *tlr2* was generated and characterised. A well-correlated expression pattern of *mtlr2* and the downstream signaling molecules *i.e., myd88, nf-kb* and *il-1β* was observed in gill, liver, spleen, intestine and kidney during the initial hours (3 to 24 hpi) of *A. hydrophila* infection in magur cat fish.

## ABBREVIATION LIST

| | |
|---|---|
| **PRR** | pathogen recognition receptor |
| ***tlr*** | toll-like receptor |
| ***myd88*** | myeloid differentiation primary response 88 |
| **cDNA** | complimentary DNA |
| **LRR** | leucine-rich repeat region |
| **TIR** | toll-interleukin 1 (*il-1*) receptor |
| **PGN** | peptidoglycan |
| **LTA** | lipoteichoic acid |
| ***nf-kb*** | nuclear Factor kappa-light-chain-enhancer of activated B |
| ***il-1β*** | interleukin 1 beta |
| **PAMP** | pathogen-associated molecular pattern |
| **DAMP** | damage-associated molecular pattern |
| **LPS** | lipopolysaccharides |

| ORF | open reading frame |
| pi | isoelectic point |
| TSB | tryptic-soy broth |
| DEPC | diethyl pyrocarbonate |

## ACKNOWLEDGEMENTS

The authors wish to express their sincere thanks to Dr. Kuldeep K. Lal, Director, ICAR-NBFGR, Lucknow, India for providing the facilities and his support, guidance and encouragement. The authors would also like to sincerely acknowledge the reviewers for their constructive remarks which have helped to improve the manuscript.

### Funding

This work was carried out with the financial in-house, ICAR-NBFGR funding support under the project code FISHNBFGRSIL201600800189. The funders had no role in study design, data collection and analysis, decision to publish, or preparation of the manuscript.

### Grant Disclosures

The following grant information was disclosed by the authors:
ICAR-NBFGR: FISHNBFGRSIL201600800189.

### Competing Interests

The authors declare that they have no competing interests.

### Author Contributions

- Chinmayee Muduli conceived and designed the experiments, performed the experiments, analyzed the data, prepared figures and/or tables, authored or reviewed drafts of the paper, and approved the final draft.
- Anutosh Paria conceived and designed the experiments, performed the experiments, authored or reviewed drafts of the paper, and approved the final draft.
- Ranjana Srivastava performed the experiments, analyzed the data, prepared figures and/or tables, and approved the final draft.
- Gaurav Rathore conceived and designed the experiments, performed the experiments, analyzed the data, authored or reviewed drafts of the paper, and approved the final draft.
- Kuldeep K. Lal conceived and designed the experiments, authored or reviewed drafts of the paper, and approved the final draft.

### Animal Ethics

The following information was supplied relating to ethical approvals (*i.e.*, approving body and any reference numbers):

Institutional Animal Ethics Committee of ICAR-NBFGR, Lucknow approved this research (NBFGR/IAEC/2019/007).
## Ethics

The following information was supplied relating to ethical approvals (*i.e.*, approving body and any reference numbers):

Institutional Research Review committee (IRC) and Research Advisory Committee (RAC).

## Data Availability

The Magur TLR2 data are available at GenBank: MT625141.

## Supplemental Information

Supplemental information for this article can be found online at http://dx.doi.org/10.7717/peerj.12411#supplemental-information.

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
