# Peer review of "Aeromonas hydrophila infection induces Toll-like receptor 2 (tlr2) and associated downstream signaling in Indian catfish, Clarias magur (Hamilton, 1822)"

_PeerJ, doi:10.7717/peerj.12411_

## Round 0.1 · original submission · Major Revisions

Please carefully consider and address comments from the reviewers, in particular please make sure that the methods provide enough detailed information. Please consider providing more data on the function of TLR.

·

Basic reporting

1. Fish gene names should be italicized, with all letters in lower-case. Protein symbols are not italicized, but only the first letter should be capitalized. All the gene and protein symbols in this manuscript need to be corrected.

2. In section 2.8, the method for primer specificity analysis is not provided, which is inconsistent with the title of this section.

3. If the authors want to keep section 3.3 as a part of Result section, more results should be provided. Currently, only the amplification efficiency of primers is shown, with no data showing the specificity of the primers (melt curve or gel image). Alternatively, section 3.3 can be removed and only Table 3 is kept and attached to section 3.4.

4. Line 85: there is a missing "a" between "play" and "vital"

5. Line 354: "higher" should be "highest"

Experimental design

1. Line 212: The authors used ef1a (italic) as the internal control for qPCR. Why is this gene selected? Although ef1a (italic) is a housekeeping gene, one cannot directly use it as internal control unless its expression level has been already shown to be unchanged by certain stress (in this case, A. hydrophila). Therefore, please provide references or experimental evidences to support it.

Validity of the findings

No comment

Additional comments

Since this is the first study of indian catfish tlr2 gene function and this gene was shown to be upregulated by A. hydrophila, it would be nice to also test whether tlr2 expression is affected by PGN, LPS, etc. As the authors mentioned (Line 369-372), the response of tlr2 to A.hydrophila might be due to these components, but it is not clear in Indian Catfish.

Reviewer 2 ·

Basic reporting

1. English should be properly scrutinized.
2. Some references are lacking (see attached PDF file with notes).
3. Article structure, figures, tables and raw data are acceptable, but should provide more data for a full-length article.

Experimental design

Experimental design is fine, but should explain why it was Gram-negative bacteria (Aeromonas hydrophila) instead of Gram-positive bacteria was used in challenge trail when most of the references indicated that TLR2 is responsible for sensing ligands from Gram-positive bacteria.

Validity of the findings

Many of the questions and suggestions are included in the PDF file uploaded.

Additional comments

More functional studies on Clarias magur TLR2 (mTLR2) is more than welcome, such as subcellular localization of mTLR2 in fish cells, protein-protein interaction of mTLR2 and potential adaptor proteins (MyD88, TIRAP, TIRF and Sarm1), and ligand specificity of mTLR2 to known TLR2 ligands (can be done using luciferase assay)

Annotated reviews are not available for download in order to protect the identity of reviewers who chose to remain anonymous.

---

## Round 0.2 · Major Revisions

Please carefully consider and address all comments from Reviewer 2, please make sure all changes are made before resubmission of the manuscript.

·

Basic reporting

No comment

Experimental design

No comment

Validity of the findings

No comment

Additional comments

The authors have addressed my concerns, and I recommend publication in the journal.

Reviewer 2 ·

Basic reporting

Figs. 1 and 2 should be corrected.

Experimental design

Good.

Validity of the findings

This is a interesting manuscript, but efforts are required to make this manuscript with a better quality.

Additional comments

The authors have made much progress on making this manuscript with a better quality for publication. However, some points are needed to be corrected.

1. PRR should not in italics and in lower case (e.g. lines 16, 17 ,19, 38 and many places in the manuscript).
2. MyD88 should not in italics and lower case in line 20 (it is the MyD88 protein to pass the signal to the downstream, not the gene itself, so do for other places not just for MyD88).
3. LRR should not in lower case (e.g. lines 26, 27). It is protein we are talking about here.
4. TIR should not in lower case (e.g. line 28).
5. TLR2, TLR1 and TLR6 should not in italics and in lower case in lines 71 and 72.
6. Fig. 1 is incorrect. Please make sure the number of LRRs are only 6 of it in mTLR2, which I found quite hard to believe (Fig. 1). Also, TLR2 should have LRR-NT. It is worth to check Quiniou et al 2013 (DOI: 10.1007/s00251-013-0694-9). The supplementary data of this paper provides each LRR for every TLRs found in channel catfish. Simply make alignment with catfish TLR2, you can easily find other LRRs that you can’t identify using SMART software.
7. Fig. 2 is incorrect. Please identify each LRR manually according to the definition of LRR, not just use online software since online software may give biased results due to software settings.
8. The authors stated that “Unlike mammal, the tlr2 in fish has extended functional attributes to recognise Gram-negative bacteria and LPS in addition to its known ligands, and Gram-positive bacteria.”. But there is no direct evidence to prove this point. Upregulation of tlr2 genes does not mean it was activated. For your information, a paper was published in 2018 (https://www.frontiersin.org/articles/10.3389/fimmu.2018.01413/full) and they show NOD1 is one of the receptor for LPS recognition. However, we can’t exclude the possibility that TLR2 might play a role in this recognition, but more evidence is needed to demonstrate this point.

---

## Round 0.3 · accepted · Accept

Thank you for making all the requested revisions.

Reviewer 2 ·

Basic reporting

The authors have responded well and have answered all the questions raised. Additionally, the corrections have been properly panted in the new version of manuscript. I think it is now ready to be process for publication.

Experimental design

no comment

Validity of the findings

no comment